# Ethosomes and Transethosomes as Cutaneous Delivery Systems for Quercetin: A Preliminary Study on Melanoma Cells

**DOI:** 10.3390/pharmaceutics14051038

**Published:** 2022-05-11

**Authors:** Francesca Ferrara, Mascia Benedusi, Maddalena Sguizzato, Rita Cortesi, Anna Baldisserotto, Raissa Buzzi, Giuseppe Valacchi, Elisabetta Esposito

**Affiliations:** 1Department of Neuroscience and Rehabilitation, University of Ferrara, I-44121 Ferrara, Italy; francesca.ferrara@unife.it (F.F.); mascia.benedusi@unife.it (M.B.); 2Department of Chemical, Pharmaceutical and Agricultural Sciences, University of Ferrara, I-44121-Ferrara, Italy or maddalena.sguizzato@unife.it (M.S.); crt@unife.it (R.C.); 3Department of Life Sciences and Biotechnology, University of Ferrara, I-44121 Ferrara, Italy; anna.baldisserotto@unife.it (A.B.); raissa.buzzi@unife.it (R.B.); 4Department of Environmental and Prevention Sciences, University of Ferrara, I-44121 Ferrara, Italy; 5Plants for Human Health Institute, Department of Animal Science, NC Research Campus Kannapolis, NC State University, Kannapolis, NC 28081, USA; 6Department of Food and Nutrition, Kyung Hee University, Seoul 130-701, Korea

**Keywords:** quercetin, ethosome, transethosome, in vitro release test, in vitro permeation test, Franz cell, wound healing, migration assay

## Abstract

The present study is aimed to design ethosomes and transethosomes for topical administration of quercetin. To overcome quercetin low bioavailability, scarce solubility and poor permeability that hamper its pharmaceutical use, the drug was loaded in ethosomes and transethosomes based on different concentrations of phosphatidylcholine. Vesicle morphology was studied by cryogenic transmission electron microscopy, while size distribution and quercetin entrapment capacity were evaluated up to 3 months, respectively, by photon correlation spectroscopy and high-performance liquid chromatography. The antioxidant property was studied by photochemiluminescence test. Quercetin release and permeation was investigated in vitro, using Franz cells associated to different membranes. In vitro assays were conducted on human keratinocytes and melanoma cells to study the behavior of quercetin-loaded nano-vesicular forms with respect to cell migration and proliferation. The results evidenced that both phosphatidylcholine concentration and quercetin affected the vesicle size. Quercetin entrapment capacity, antioxidant activity and size stability were controlled using transethosomes produced by the highest amount of phosphatidylcholine. In vitro permeation studies revealed an enhancement of quercetin permeation in the case of transethosomes with respect to ethosomes. Notably, scratch wound and migration assays suggested the potential of quercetin loaded-transethosomes as adjuvant strategy for skin conditions.

## 1. Introduction

Quercetin (3,3′,4′,5,7-pentahydroxyl-flavone, QT) is a natural flavonoid occurring in a wide variety of vegetables, flowers and fruits [1]. QT anti-inflammatory, antioxidant, antiviral, antimicrobial, and anticarcinogenic properties make it the object of several studies aimed to employ the drug in pharmaceutics, and in nutraceutics [2,3,4,5]. Moreover, different studies evidenced QT potential to treat skin pathologies, including skin cancer [6].

Melanoma can be considered as the most aggressive skin tumor, despite early detection, appropriate surgical resection and adjuvant therapy [7]. Since some drugs approved by FDA to treat melanoma (i.e., dacarbazine, hydroxyurea, and interleukin-2) can induce resistance and toxicity, there is an unmet need of new biological therapeutics to be employed as combination treatments [8,9]. In this regard, recently QT has been proposed in the prevention and treatment of melanoma, mainly in reason of its inhibitory effect against tyrosinase activity [10]. However, some drawbacks limit the pharmaceutical use of QT, such as poor water solubility (≈10 μg/mL), scarce bioavailability, and chemical instability [11]. In addition the topical treatment of cutaneous pathologies using QT can be hampered by its low cutaneous permeability [12]. 

In order to overcome these issues, recently technologically advanced delivery systems were designed for QT topical administration, such as microemulsions, lipid microparticles, liposomes, and ethosomes (ETO) [12,13,14,15]. ETO are phospholipid vesicular systems usually constituted of phosphatidylcholine (PC), water and ethanol (20–45% *v*/*v*) [16]. These vesicular systems can be defined as the second generation of liposomes, being characterized by higher malleability, longer stability and better capability to entrap insoluble molecules [17]. Indeed, in ETO, the presence of PC and ethanol exerts a penetration enhancement effect, due to a disorganization of the *stratum corneum* barrier, leading to an improvement of drug permeability through the skin [18,19]. Moreover, in order to confer to ETO vesicles an even more flexible structure, the so-called transethosomes (T-ETO) can be formulated by the addition of non-ionic surfactants, as well as of polyethylene glycol, to PC ethanol solution [20,21,22]. In T-ETO the presence of surfactants, employed as “edge activators”, should further improve the vesicle transdermal potential, making them a novel and promising technology for the treatment of different cutaneous conditions [23,24,25]. Indeed ETO and T-ETO can enhance the transdermal delivery of the entrapped drugs, since the vesicles can pass through the stratum corneum barrier, promoting the drug passage towards the dermis [19,20]. Nonetheless, the possibility to treat melanoma by topical application of QT loaded ETO and T-ETO has never been explored.

Thus, in this study firstly ETO and T-ETO were designed and compared for QT loading, then their transdermal potential and suitability as possible candidates in the adjuvant therapy of melanoma have been investigated.

An important aspect in the development of topical delivery system is the evaluation of the in vitro diffusion kinetics of the loaded drug, especially to compare the influence of composition in the case of different vehicles, providing key insights into the relationships between drug and formulation [26]. Particularly in vitro release tests (IVRT) are predictive of the diffusion of drug from the product to the skin surface, while in vitro permeation tests (IVPT) are employed to study drug diffusion from the product matrix to the skin and systemic circulation [27,28]. To this aim, diffusion cells can be employed, associated to synthetic membranes, that should not be rate-limiting to drug release, acting only as a barrier to separate the formulation from the receptor medium [27]. In this respect the choice of membranes is a topic of concern. In the case of IVRT, usually synthetic membranes based on nylon, mixed cellulose esters, or polytetrafluoroethylene are used. With regard to IVPT, despite natural membranes can achieve predictable results regarding in vivo drug permeation, many factors restrict their employment, such as variability of results, scarce availability, and high costs [27,28,29,30,31]. To overcome these drawbacks, recently peculiar synthetic membrane systems, able to reproduce the peculiar *stratum corneum* assembly, have been proposed [31]. Particularly, the multiple polymeric membrane system named Strat-M^®^, based on two polyether sulfone layers overlapped to one polyolefin bottom layer, has skin-like tortuous porous structure [32]. Moreover, the impregnation with synthetic lipids further confers to Strat-M^®^ a resemblance with the skin, characterized by hydrophilic and lipophilic compartments, providing barrier properties [32].

Therefore, in this study the diffusion performances of QT loaded-ETO and T-ETO were evaluated by Franz cell, using Nylon (NY) and polytetrafluoroethylene (PTFE) membranes for IVRT, while, for IVPT, both STRAT-M^®^ and human *stratum corneum*-epidermis (SCE) have been employed.

Remarkably, in the present investigation a preliminary biological study was conducted to explore the feasibility of QT loaded-ETO and T-ETO for improving keratinocyte healing performance and possibly as a melanoma adjuvant treatment, by performing scratch wound healing and migration assays on normal human keratinocytes and melanoma cells.

## 2. Materials and Methods

### 2.1. Materials

Quercetin (3,3′,4′,5,6-Pentahydroxyflavone,2-(3,4-Dihydroxyphenyl)-3,5,7-trihydroxy-4H-1-benzopyran-4-one, QT) and polysorbate 80 (polyoxyethylenesorbitan monooleate, tween 80, T80), were purchased from Sigma-Aldrich (St. Louis, MO, USA). The soybean lecithin (PC) (90% phosphatidylcholine) was Epikuron 200 from Lucas Meyer (Hamburg, Germany). Nylon (New York, NY, USA), polytetrafluoroethylene (PTFE, Whatman^®^, Maidstone, UK) (pore size 200 nm), and STRAT-M^®^ membranes were purchased from Sigma Aldrich (Milan, Italy). Solvents were of HPLC grade and all other chemicals were of analytical grade.

### 2.2. Ethosome Preparation 

ETO preparation was obtained by a cold method based on dropwise addition of bidistilled water to PC ethanol solutions [20]. Briefly, PC was previously solubilized in ethanol (30 mg/mL) under stirring at 750 rpm (IKA RCT basic, IKA^®^-Werke GmbH and Co. KG, Staufen, Germany). After complete solubilization, water was slowly added to the PC solution up to a final 70:30 (*v*/*v*) water/ ethanol ratio. The magnetic stirring was maintained for 30 min. To prepare QT-loaded ETO, the drug (0.5 mg/mL) was solubilized in the PC ethanol solution before the addition of water. 

### 2.3. Transethosome Preparation 

In the case of T-ETO, the surfactant T80 was solubilized in PC ethanol solutions (30 or 90 mg/mL), up to a final 1% *w*/*v* concentration. Afterwards T-ETO preparation was conducted with the same above reported procedure employed for ETO, resulting in a final T80 0.3% *w*/*w* concentration. To prepare QT-loaded T-ETO, the drug (0.5 mg/mL) was solubilized in the PC/T80 ethanol solution before the addition of water.

### 2.4. Cryo-Transmission Electron Microscopy (Cryo-TEM)

For cryo-TEM analyses samples were vitrified following a method previously reported [20]. Namely, a 2 μL aliquot of sample was put for few seconds on a lacey carbon filmed copper grid (Science Services, München, Germany). After removing most of the liquid by a blotting paper, a thin film stretched over the lace holes was obtained. Vitrification was achieved by rapid immersion of specimen into liquid ethane cooled to approximately 90 K (−180 °C) by liquid nitrogen in a temperature-controlled freezing unit (Leica EMGP, Leica, Wetzlar, Germany). The sample preparation procedure was conducted at controlled constant temperature in the Leica EMGP chamber. The vitrified specimen was transferred to a Zeiss/Leo EM922 Omega EFTEM (Zeiss Microscopy GmbH, Jena, Germany) transmission electron microscope using a cryoholder (CT3500, Gatan, Munich, Germany). During the microscopy observations, sample temperature was kept below 100 K. Specimens were examined with reduced doses ≈1000–2000 e/nm^2^ at 200 kV. Zero-loss filtered images (ΔE = 0 eV) were recorded by a CCD digital camera (Ultrascan 1000, Gatan, Munich, Germany) and analyzed by a GMS 1.9 software (Gatan, Munich, Germany). 

### 2.5. Photon Correlation Spectroscopy (PCS)

Vesicle size distribution was measured using a Zetasizer Nano-S90 (Malvern Instr., Malvern, UK) with a 5 mW helium neon laser and a wavelength output of 633 nm. Measurements were performed at 25 °C at a 90° angle and a run time of at least 180 s. Samples were diluted with bidistilled water in a 1:20 *v*/*v* ratio. Data were analyzed using the “CONTIN” method [33]. Measurements were performed thrice for 3 months after nano-vesicular forms production, calculating the mean ± standard deviation (s.d.). The statistical differences between Z Average mean diameters 1 and 90 days after sample preparation were evaluated by t student test, GraphPad Prism 9 software (GraphPad Software Inc., San Diego, CA, USA), considering values of *p* < 0.05 as statistically significant.

### 2.6. QT Entrapment Capacity Evaluation

Briefly, to evaluate the amount of drug associated to ETO and T-ETO vesicles, QT content was determined the day after preparation by ultrafiltration using a centrifugal filter device (Microcon centrifugal filter unit YM-10 membrane, NMWCO 10 kDa, Sigma-Aldrich, St. Louis, MO, USA) and HPLC analysis. Namely, 500 μL of QT-loaded ETO and T-ETO were poured in the sample reservoir part of the device and subjected to ultrafiltration (Spectrafuge™ 24D Digital Microcentrifuge, Woodbridge, NJ, USA) at 8000 rpm for 20 min. Afterwards, both retentate and filtrate fractions were withdrawn, respectively, from the sample reservoir part or the vial, and diluted with ethanol (1:10, *v*/*v*). Before HPLC analysis, the diluted retentate was stirred for 30 min and filtered by nylon syringe membranes (0.22 μm pore diameter), while the filtrate fraction was analyzed as such. The entrapment capacity (EC) was determined as follows:EC = QT/T_QT_ × 100(1)
where QT is the amount of drug retained by the vesicles and T_QT_ is the total content of QT employed for ETO and T-ETO preparation. In addition, for chemical stability studies, QT content was evaluated monthly for 3 months on the whole nano-vesicular dispersion stored at 22 °C. Briefly 100 μL of dispersion were diluted with ethanol in a 1:10 *w*/*w* ratio, stirred for 30 min, and filtered by nylon syringe filters (0.22 μm pores). Afterwards QT content was evaluated by HPLC as below reported.

### 2.7. Franz Cell Diffusion Experiments

Franz vertical cells (orifice diameter 0.9 cm; PermeGear Inc. Hellertown, PA, USA) were employed for IVRT and for IVPT. Notably NY and PTFE membranes were used for IVRT, while STRAT-M^®^, and human stratum corneum-epidermis (SCE) membranes were employed for IVPT [32]. SCE membranes, obtained from breast reduction operations, were dried in a desiccator at ~25% relative humidity, wrapped in aluminum foil and stored at 4 ± 1 °C until use. In order to detect the SCE integrity and thickness, portions of membranes were mounted on metal stubs using a plastic conductive carbon cement, coated with a gold sputter coating Edwards S150 (Edwards High Vacuum, Burgess Hill, UK) and observed under a Scanning Electron Microscope (SEM) ZeissEVO 40 (Carl Zeiss AG, Jena, Germany). 8 S1 summarizes the membrane features.

Both for IVRT and IVPT, samples of dried membranes were rehydrated by immersion in ethanol/water 50:50, *v*/*v* for 1 h, before assembling in Franz-type diffusion cells. The receptor compartment of the cell contained 5 mL of ethanol:water 50:50, *v*/*v* in order to assure sink conditions, stirred at 500 rpm by a magnetic bar and thermostated at 32 ± 1 °C during the experiments [27]. Five hundred microliters of QT-loaded ETO and T-ETO, or QT ethanolic solution (ethanol:water 30:70, *v*/*v*) (QT 0.5 mg/mL) (SOL-QT) were placed on the membrane surface in the donor compartment that was afterwards sealed to avoid evaporation. At predetermined time intervals comprised between 1 and 24 h, samples (0.5 mL) of receptor phase were withdrawn and analyzed by HPLC to evaluate the QT content. Each removed sample was replaced with an equal volume of simple receptor phase. The QT concentrations were determined 6–12 times in independent experiments, the mean values ± s.d. were calculated. By the end of the Franz cell experiment, the membranes were cut, put in vials containing 1 mL of ethanol and subjected to ultrasonication (Branson, Bransonic^®^ M Mechanical Bath 3800, Emerson, St. Louis, MO 63136, USA) for 15 min. Afterwards the suspension was filtered by nylon syringe membranes and analyzed by HPLC. The amount of QT associated to the membranes (M_QT_) was calculated.

#### 2.7.1. In Vitro Release Test (IVRT)

For data analysis, in the case of IVRT, QT amount released (μg/cm^2^) was plotted as a function of the square root of time. To compare the QT release kinetics from the different dispersions, the following parameters were evaluated: “R_QT_” the slope of the cumulative amount of QT released versus the square root of time; lag-time “T_lag_” extrapolated from the intercept of the release profile with x-axis; and “A_QT_” the cumulative amount of QT released at the last sampling time (8 h). Furthermore, the mechanism of QT release from the vehicles was studied applying, for in vitro release data, regression analysis of the cumulative %QT release vs. time (zero order kinetic model), the log cumulative of %QT remaining vs. time (first order kinetic model), and the cumulative %QT release vs. square root of time (Higuchi model). Model fit was performed by means of the DDsolver add-in Excel module [34].

#### 2.7.2. In Vitro Skin Permeation Test (IVPT)

In the case of IVPT, to analyze data, QT amount permeated (μg/cm^2^) was plotted as a function of time. Fick’s law was considered since it describes the steady-state permeation through the skin, assuming that, under sink conditions, drug concentration in the receptor compartment is negligible with respect to that in the donor compartment [35]. The steady-state flux of drug per unit area “Jss” is described as:Jss = P × Cd × D/e(2)
where P is the partition coefficient, Cd is the drug concentration in the donor compartment, D is QT diffusion coefficient, and e is the thickness given by the supplier or measured. Accordingly, QT permeability coefficients “Kp” and T_lag_ values were calculated considering the steady-state portion of QT cumulative penetration profiles versus time. The slope of the linear part of the curves yielded the pseudo steady-state flux “Jss” (µg/cm^2^/h) [35]. Kp was calculated according to Equation (3):Kp = Jss/Cd(3)

The D value was calculated from T_lag_ according to Equation (4):T_lag_ = e^2^/6 × D(4)

At last, P was calculated considering the Equation (5):Kp = D × P/e(5)

### 2.8. HPLC Analysis

HPLC analyses were performed using Perkin Elmer, Series 200 HPLC Systems equipped with a micro-pump, an auto sampler, and an UV-detector operating at 255 nm. A stainless-steel C-18 reverse-phase column (15 × 0.46 cm) packed with 5 μm particles (Hypersil BDS C18 Thermo Fisher Scientific S.p.A., Milan, Italy) was eluted at a flow rate of 1 mL/min with a mobile phase containing acetonitrile/water 40:60 *v*/*v*. Injection volume and retention time were 5 μL and 2.8 min, respectively.

### 2.9. Photochemiluminescence (PCL) Test

The PCL test is based on the reaction of a specific radical species, the superoxide anion (O_2_^•−^), photochemically generated by UV radiation, with a compound capable of emitting chemiluminescence, according to the method of Popov and Lewin [36]. The marker used was luminol, a molecule that, oxidized by free radicals, emits light that can be measured with a special instrument (Photochem^®^, Analytik Jena, Leipzig, Germany). The presence of antioxidant substances in the mixture of the ration deactivates the radical species by inhibiting the emission of chemiluminescence. PCL analysis is very fast and sensitive. Indeed, the application of two different analytical protocols, named ACW (antioxidant capacity of water soluble compounds) and ACL (antioxidant capacity of lipid soluble compounds) enables to evaluate the contributions to the total antioxidant capacity of both the water-soluble molecules (e.g., vitamin C and amino acids) and the fat-soluble one (e.g., tocopherols, tocotrienols, carotenoids). The kinetic light emission curve, which exhibits no lag phase in ACL studies, was monitored for 180 s and expressed as micromoles of Trolox (standard) per gram of sample. The areas under the curves were calculated using the PCL soft control and analysis software. As greater concentrations of Trolox working solutions were added to the assay medium, a marked reduction in the magnitude of the PCL signal and hence the area calculated from the integral was observed. This inhibition was used as a parameter for quantification and related to the decrease in the integral of PCL intensities caused by varying concentrations of Trolox. The observed inhibition of the signal was plotted against the concentration of Trolox added to the assay medium. The concentration of the added extract solution was such that the generated luminescence during the 180 s sampling interval fell within the limits of the standard curve. At least two blank-measurements should be carried out (only reagents of ACL-kit without sample and/or Trolox): 2.3 mL of ACL-Diluent, 0.2 mL of Reaction buffer, and 0.025 mL of Luminol Stock solution. Samples of ETO, T-ETO, SOL-QT, QT-loaded ETO, and QT-loaded T-ETO were suitably diluted in methanol prior to analysis. The antioxidant assay was carried out in triplicate for each sample, and 10 µL of the diluted samples (about 300–375 µg) in HPLC-grade methanol were sufficient to correspond to the standard curve.

### 2.10. Biological Activity Studies

#### 2.10.1. Cytotoxicity Study

HaCaT cells and HT-144 melanoma cells were cultured in Dulbecco’s modified Eagle’s medium High Glucose (DMEM), (Lonza, Milan, Italy), supplemented with 10% FBS (foetal bovine serum), 100 U/mL penicillin, 100 μg/mL streptomycin and 2 mM L-glutamine. Cells were incubated at 37 °C for 24 h in 95% air/5% CO_2_ until 80% confluence.

For cell treatments, the different vehicles were initially dissolved in cell culture medium to obtain the stock solutions containing QT 500 µg/mL, then further diluted to reach final concentrations of QT 0.5–5 μg/mL. 

First, the two cell lines were grown in 96-well plates at a density of 2 × 10^4^ cells/well in 200 μL of media for MTT assay. Seeded cells were exposed to unloaded and QT-loaded formulations at various QT concentrations, ranging from 0.5 μg/mL to 5 μg/mL for 24 h. After complete removal of the treatment to avoid any color interference, 50 μL of serum-free media and 50 μL MTT (0.5 mg/mL) were added and incubated for 3 h. The insoluble purple formazan crystals were then dissolved in 100 μL of DMSO at 37 °C for 15 min. After shaking, the solution absorbance was measured with a spectrophotometer at 590 nm, using 670 nm as a reference wavelength, and, thus, converted into a percentage of viability [37].

#### 2.10.2. Wound Healing Assay

HaCaT cells and HT-144 melanoma cells were grown in complete DMEM medium to confluent monolayer on 24-well plates and then were mechanically scratched with a 200 μL sterile pipette tip. Cellular debris were removed by washing off with PBS and based on the cytotoxicity data cells were immediately treated with 2.5 μg/mL of QT-loaded and unloaded formulations, followed by incubation at 37 °C. Images of the scratches for each sample were recorded at different time points (i.e., 0, 12, 24 h post- scratch). The wound healing rate was analyzed with ImageJ software (National Institutes of Health, Bethesda, MD, USA) and compared to the wound area at T_0_ [38,39]_._

#### 2.10.3. Cell Migration Assay 

HaCaT cells were seeded in 10 cm^2^ petri dishes and pretreated with 2.5 μg/mL of QT-loaded and unloaded formulations for 24 h; then, 100,000 cells were resuspended in 400 μL of serum-free mediua and seeded in 8 μm pore size transwells (Falcon^®^ Permeable Support for 12-well Plate with 8.0 µm Transparent PET Membrane, Sterile) coated with 0.15 mg/mL bovine collagen IV. After 30 min, 1100 μL of complete media was added at the bottom of each well, acting as a chemoattractant. Twenty-four hours later the transwell inserts were fixed for 10 min with 70% ethanol, stained with 0.02% of Coomassie Blue for 15 min, and rinsed with double-distilled water. HaCaT cells left unmigrated in the upper part of the transwell were gently removed with a cotton swab, and pictures of 5 randomly selected fields were acquired after 24 h (magnification 20×). 

Automated quantification of the migrated cells was performed using ImageJ program and the number of migrated cells present in five fields/well was counted. Data were reported as percentage of migrated cells.

For HT-144 melanoma cells the same protocol was applied, except that the cells were detached and treated with 2.5 μg/mL of QT-loaded, or unloaded forms immediately before the seeding in 8 μm pore size transwells. The percentage of migrated cells was then evaluated after 24 h, as above described [39].

### 2.11. Statistical Analysis

For each variable tested the analysis of variance (two-way ANOVA), followed by Tukey’s post-hoc test, was used. A probability of less than 0.05 was considered significant. Data are expressed as mean ± s.d. of duplicate determinations obtained in at least 3 independent experiments.

## 3. Results

### 3.1. Preparation of Ethosomes and Transethosomes

Lipid vesicular systems constituted of PC were designed as topical vehicles for QT solubilization and application on the skin. Particularly, due to QT solubility in ethanol (2 mg/mL), ETO and T-ETO were produced, using ethanolic solutions of PC, as the main vesicle component (Table 1) [40]. ETO and T-ETO were easily formed upon addition of water to PC ethanolic solution under stirring. Particularly, based on our previous studies, PC 0.9%, and ethanol 30% were selected (Table 1), resulting in milky homogeneous dispersions (ETO_0.9_) [41]. With respect to other studies conducted on QT-loaded ETO [13,14], in the present study the ETO composition was improved by adding to the PC ethanolic solution the non-ionic surfactant T80, studying the effect of different PC concentrations, possibly stabilizing the vesicles, while increasing their solubilizing power and transdermal efficacy. The resulting T-ETO_0.9_ dispersions appeared homogeneous and characterized by a more translucent aspect with respect to ETO. Notably, the T80 concentration was 0.3% *w*/*w*, being previously selected as suitable to obtain stable and homogeneously sized vesicles [20,41]. Since research studies have demonstrated that drug entrapment efficiency is influenced by the phospholipid concentration of vesicles [17], the PC concentration of T-ETO has been increased, up to 2.7% *w*/*w* (3-fold). The resulting T-ETO_2.7_ were homogeneous and quite translucent. It should be underlined that ETO produced by PC 2.7% *w*/*w*, in the absence of T80, underwent sedimentation and phase separation [20]. QT-loaded ETO and T-ETO, obtained by previous QT solubilization in PC ethanol solutions, were characterized by a yellowish color and homogeneous aspect, free from separation phenomena.

### 3.2. Characterization of Ethosomes and Transethosomes

#### 3.2.1. Morphology and Size

To gain information about the morphology and size of ETO and T-ETO, cryo-TEM and PCS were employed. Figure 1 reports cryo-TEM images of ETO_0.9_-QT, T-ETO_0.9_-QT, and T-ETO_2.7_-QT.

In all cases, the PC double layer organization, gave rise to multilamellar and unilamellar spherical or elongated vesicles. The presence of T80 slightly affected the vesicle structures. Indeed, in the case of T-ETO_0.9_-QT, unilamellar vesicles and oligolamellar ones were detectable (Figure 1b), while in the case of T-ETO_2.7_-QT (Figure 1c), multivesicular vesicles together with multilamellar vesicles were detected. 

As reported in Table 2, mean diameters, expressed as Z Average, measured 1 day after preparation, ranged between 160 and 400 nm, with notable differences between the vesicles based on PC 0.9 or 2.7% *w/w*. 

In general, smaller mean diameters were obtained in the case of ETO and T-ETO produced by the lower PC concentration. The presence of T80 led to a 50 nm decrease of mean diameter for T-ETO_0.9_, with respect to ETO_0.9._ Conversely, in the case of T-ETO_2.7,_ as expected, the increase of PC doubled up the vesicle mean diameter [13], while QT loading affected Z Average of all kinds of vesicles, especially in the case of T-ETO_2.7_-QT (Table 2)_._ The dispersity indexes below 0.3, indicate homogeneous size distributions, mostly characterized by the presence of one peak. Nonetheless, in the case of T-ETO_0.9_-QT, T-ETO_2.7_, and T-ETO_2.7_-QT, PCS analyses evidenced bimodal size distributions with less represented peaks (area 8–13%) measuring ≈ 1000 nm.

#### 3.2.2. QT Entrapment Capacity (EC)

To evaluate the EC of QT in ETO_0.9_-QT, T-ETO_0.9_-QT, and T-ETO_2.7_-QT, the lipid phase was separated from the aqueous one by ultrafiltration, before dissolution in ethanol in order to promote vesicle disaggregation. Table 3 reports the EC results. The quantification of QT in both phases confirmed the total recovery of the drug in the dispersion, although an aliquot of QT (36–43%) was detected in the water phase. The highest EC was obtained in the case of T-ETO_2.7_-QT.

### 3.3. Evaluation of QT Antioxidant Activity

In order to compare the effect of QT entrapment in ETO and T-ETO, their antioxidant activity was evaluated by PCL test. The results are summarized in Table 3 and expressed as ACL. The ACL values of ETO_0.9,_ T-ETO_0.9,_ and T-ETO_2.7_ were not reported, being irrelevant, suggesting that empty vesicles did not possess an intrinsic antioxidant activity.

Conversely, QT-loaded vesicles showed an antioxidant activity, remarkably improved in the case of T-ETO_2.7_-QT. Indeed, ETO_0.9_-QT and T-ETO_0.9_-QT possess a similar antioxidant capacity (differences not statistically significant), lower with respect to QT ethanolic solution (SOL-QT), while T-ETO_2.7_-QT displayed the highest antioxidant capacity, being 1.32-fold higher with respect to SOL-QT and 1.6-fold higher with respect to T-ETO_0.9_-QT (ACL values extremely significantly different; *p* < 0.0001). 

### 3.4. Stability Studies

The effect of storage on vesicle size and EC of QT was investigated on nano-vesicular forms kept at 22 °C for 3 months (Figure 2). 

Vesicle size was quite stable, apart from T-ETO produced by the lowest PC amount. Particularly, in the case of T-ETO_0.9_-QT, vesicle mean diameters underwent a 1.6-fold increase. The increase of vesicle size was statistically significant, both for T-ETO and T-ETO_0.9_-QT. Conversely, the size of T-ETO_2.7_-QT was quite stable (14 nm increase), while displaying the largest Z Average diameters (Figure 2a). To investigate the nano-vesicular system capability to control QT chemical stability, the variation of the total QT residual content was compared. As represented in Figure 2b, despite QT content decreased within 3 months in all samples, T-ETO_2.7_-QT and T-ETO_0.9_-QT better controlled QT degradation with respect to ETO_0.9_-QT, displaying, respectively, 68, 65 and 55% residual content of QT. 

### 3.5. IVRT 

An in vitro Franz cell system was employed to compare the ability of the different vesicular systems to control QT release. IVRT is employed in the development of topical drug delivery systems as a screening tool to compare the performance characteristics of several prototypes formulations [27,28]. Particularly, different types of membranes were considered, i.e., the hydrophilic NY, and the hydrophobic PTFE (Appendix A) with the aim to investigate the membrane effect on QT release. Figure 3 shows the release profiles of QT from ETO_0.9_-QT, T-ETO_0.9_-QT, T-ETO_2.7_-QT and SOL-QT obtained using NY (panel a) and PTFE (panel b) membranes. 

The slopes of QT release were less steep in the case of ETO_0.9_-QT, T-ETO_0.9_-QT, T-ETO_2.7_-QT with respect to SOL-QT (differences statistically significant, *p* < 0.0001), indicating that QT entrapment in the vesicle can control the drug release. As clearly appreciable, the type of membrane affected the release kinetics of QT. QT release rate R (corresponding to the slope of the profiles) are reported in Table 4. Notably, in the case of the hydrophilic NY membrane, in general R values were lower with respect to the hydrophobic PTFE. Particularly, in the case of T-ETO_2.7_-QT, R_QT_ value was 3.33-fold higher using PTFE rather than NY. 

Indeed, using the hydrophilic NY, the R_QT_ values follow the order SOL-QT > T-ETO_0.9_-QT > ETO_0.9_-QT > T-ETO_2.7_-QT, whilst in the case of the hydrophobic PTFE the R_QT_ order was SOL-QT > T-ETO_2.7_-QT > T-ETO_0.9_-QT > ETO_0.9_-QT, suggesting that NY membrane did not represent an inherent barrier but in some extent could influence the release of QT. This hypothesis is corroborated by the longer T_lag_ values found in the case of NY membranes, the lower amounts of QT released by the end of IVRT (A_QT_), as well as by the remarkably higher amount of residual QT (M_QT_) associated to NY membrane by the end of the experiments. 

Since NY seemed to exert a rate-limiting effect, PTFE would appear more suitable for QT release study. Therefore, to gain insight on the mechanism of QT release from the nanovesicle systems through NY and PTFE membranes, QT release amounts were fitted to different kinetic models [42]. The parameters of the equations describing the different models of order (R^2^, m and c) are reported in Appendix A. The kinetic model showing higher correlation coefficient (R^2^) accounts for the best-followed model. According to this criterium, ETO_0.9_-QT, T-ETO_2.7_-QT and T-ETO_2.7_-QT release profiles well fitted zero-order models, both in the case of NY and PTFE membranes (R^2^ 0.98–0.99). A pharmaceutical dosage form following this model releases the same amount of drug by unit of time, independently of its concentration, suggesting a QT sustained release by the vesicles acting as reservoir, possible resulting in a pharmacological prolonged action [43].

### 3.6. IVPT 

To investigate the influence of vesicular systems on QT permeation, Franz cell was associated to the synthetic membrane STRAT-M^®^ or to the natural SCE. The drug permeation process from the vehicle into the skin or membrane layers can be described as a passive kinetic process comprising two main steps [35]. The first limiting step, corresponding to drug release from its vehicle, can be evaluated by the partition coefficient (P), which reflects the favorite distribution of the drug in the skin/membrane or the vehicle. The second limiting step, represented by the drug diffusion through the lipophilic and hydrophilic layers of the skin/membrane, depends on the physico-chemical properties of the drug and on the vehicle composition. The permeation parameters related to this latest step are the diffusion coefficient (D) and the permeability coefficient (Kp). 

The permeation profiles of QT through Strat-M^®^ for all formulations are shown in Figure 4a,b while fluxes (Jss) and other permeation parameters are reported in Appendix A. 

The fastest QT kinetic was detected in the case of T-ETO_0.9_-QT, followed by SOL-QT, T-ETO_2.7_-QT and ETO_0.9_-QT, (Figure 4a). As reported in Appendix A, *p* value of T-ETO_0.9_-QT was 4-fold higher with respect to ETO_0.9_-QT, while the lowest *p* value was found in the case of T-ETO_2.7_-QT. Conversely, the highest Kp, D, as well as amount of QT diffused after 24 h (A_QT_ total) were achieved by T-ETO_0.9_-QT. The differences between the Kp values of SOL-QT vs. vesicular systems were statistically significant, apart from T-ETO_2.7_-QT. 

To confirm the behavior of QT-loaded vesicular systems applied on the skin, human SCE membranes were employed, since the *stratum corneum* is the primary barrier to the drug passage from the external surface into the living epidermis. This non-viable compacted desquamate structure represents a highly suitable tissue for in vitro Franz cell studies [44]. SEM analysis of SCE samples enabled to control the membrane integrity and thickness before IVPT (Appendix A). Remarkably, the QT permeability profiles obtained using SCE reflect the trend found by STRAT-M^®^ (Figure 4c,d). Appendix A reports IVPT parameters obtained using the SCE membrane.

It is noteworthy that SOL-QT displayed the lowest Kp value (10-fold and 6-fold lower with respect to T-ETO and ETO, respectively), while the highest Kp was achieved by T-ETO_0.9_-QT, followed by T-ETO_2.7_-QT. Statistical differences between Kp values extremely significant (*p* < 0.0001). 

### 3.7. Biological Activity Studies

Based on encouraging results indicating the ability of ETO and T-ETO to control QT stability and to improve its permeability, the formulations were further investigated regarding their in vitro biological activity. In addition to the largely described antioxidant and anti-inflammatory activity of QT related to its capacity to inhibit the reactive oxygen species (ROS) production [3,4,5], the drug has been reported to exert a potent anti-cancer effect, due to its capability to decrease proliferation, migration, invasion, and to stimulate apoptosis [6,45]. Remarkably, since different studies demonstrated the QT activity against melanoma [15,46,47,48], to gain preliminary information on the potential of QT-loaded nano-vesicular forms in the treatment of melanoma, wound healing and migration assays were performed on human keratinocytes and HT-144 melanoma cells.

#### 3.7.1. Cell Viability

As preliminary screening, cytotoxicity of QT-loaded and unloaded forms was determined by MTT assay. HaCaT cells and HT-144 melanoma cells were treated for 24 h with different amounts of QT-loaded and unloaded forms (0.5, 1, 2.5 or 5 μg/mL). In both cell lines a significant decrease in cell viability was detected only for 5 μg/mL of QT (≈50%), compared to untreated cells (Appendix A). Based on these results, we selected 2.5 μg/mL of QT-loaded and unloaded formulations as the highest non-cytotoxic concentration for the subsequent assays, since it did not induce a decrease in cell viability higher than 35%.

#### 3.7.2. Wound Healing Assay

To evaluate the possible role of QT-loaded and unloaded nano-vesicular forms on cell migration, scratch wound healing assay was performed in both cells’ lines. As depicted in Figure 5a,b, HaCaT cells treated with ETO_0.9_-QT, T-ETO_0.9_-QT and T-ETO_2.7_-QT showed a delayed wound closure compared to the control cells. 

As depicted in Figure 5a, left panel, at 12 h time point, for all the formulations, the wound was still 90% open, compared to the control (60%) and the unloaded formulations (circa 65%). At later time points, 24 h, (Figure 5a right panel) some differences among the applied formulations were observed. Indeed, the results suggest that ETO_0.9_-QT and T-ETO_2.7_-QT had a less effective wound closure response (~75% and ~70%, respectively) compared to the control condition (30% open) or to the unloaded formulations (50% open). HT-144 melanoma cells (Figure 6a,b) at 12 h time point showed a trend similar to that observed in the case of HaCaT cells, with a delayed wound closure for all QT loaded nano-vesicular forms. Although similar response was also observed at 24 h time point (Figure 6a), it should be mentioned that in this case QT-loaded nano-vesicular forms enabled a more efficient wound closure effect with respect to the HaCaT cells. In particular, HT-144 cells treated for 24 h (right panel) with ETO_0,9_-QT or T-ETO_2,7_-QT showed a significant delay in wound closure, as demonstrated by the wound closure quantification (Figure 6b).

#### 3.7.3. Migration Assay

Considering that the scratch-wound assay is not able to discriminate between a proliferative and migratory effect, the transwell migration assay was performed to better elucidate the possible role of QT-loaded nano-vesicular systems on cell migration. As depicted in Figure 7, no effects were noticed when HaCaT cells were treated with QT-unloaded nano-vesicular forms, excluding, therefore, any possible migratory effect due to the unloaded formulations.

Conversely, after 24 h of pretreatment with ETO_0.9_-QT, T-ETO_0.9_-QT and T-ETO_2.7_-QT, a significant and dramatic reduction of cellular migration was observed. This response was particularly significant in the case of T-ETO_2.7_-QT, when compared to control condition or to unloaded formulations, confirming the delayed wound closure results reported in Figure 6. Based on this evidence, in order to evaluate a possible role of QT-loaded nano-vesicular forms in the treatment of melanoma, their effects on HT-144 melanoma cells migration were analyzed (Figure 8).

Furthermore, in this case no differences were observed between control condition and cells treated with ETO_0.9_, T-ETO_0.9_ and T-ETO_2.7_, while in the case of ETO_0.9_-QT, T-ETO_0.9_-QT and T-ETO_2.7_-QT, a significant decrease of the melanoma cell migratory capabilities was observed (~70%, 80% and 90%, respectively). 

## 4. Discussion

This study aimed to design an efficacious transdermal vehicle for QT, PC based nanosystems. Indeed the peculiar amphiphilic structure of PC allows to (i) spontaneously form different lyotropic crystalline phases in the presence of water, such as micelles and vesicles, (ii) solubilize lipophilic or amphiphilic molecules, (iii) confer penetration enhancer properties, due to its chemical similarity with stratum-corneum lipids [49,50,51]. It is noteworthy that the preparation method, based on the simple addition of water to PC ethanol solution under magnetic stirring, is particularly advantageous with respect to other production protocols described for ETO preparation [12,13,14]. Indeed, the method avoids the use of toxic organic solvents, does not require high temperature, preventing possible degradation of thermolabile drugs, it is cost effective and eco-sustainable, since it consumes low energy, and moreover it is easy to be scaled-up. 

With regard to the different lamellar organization of ETO and T-ETO observed by cryo-TEM, it should be mentioned that a disordering of the lamellar organization can be the result of the intercalation of the T80 oleate chain within the lipid bilayer. In addition, the polar terminal groups of the polyoxyethylene oxide chains could affect the aqueous compartment swelling [52].

Concerning PCS data, the increase of vesicle size exerted by QT could be ascribed to its positioning near the polar head groups of the lipids, at the interface between the glycerol groups and the apolar lipid chains, increasing the average distance among PC molecules constituting the vesicle bilayer, as recently suggested by Eid et al. [53]. A graphical representation of ETO_0.9_-QT vs. unloaded ETO_0.9_ is reported in Appendix A.

It should be noted that ETO and T-ETO are characterized by a peculiar elasticity due the presence of ethanol, that makes vesicles malleable, thus enhancing their penetration [16,17,18]. In this respect, otherwise than solid nanoparticles, in the case of ETO and T-ETO, the vesicle size scarcely affects their passage through the biological membranes. Moreover the presence of ethanol can induce disordering effects on the membranes of cells, such as keratinocytes, thus promoting ETO and T-ETO entrance, as previously demonstrated [20].

The quantification of QT in both dispersed and dispersing phases confirmed the total recovery of the drug in the dispersion, denoting that the preparation strategy avoided drug loss on mechanical devices and preserved QT from possible thermal or light degradation. After ultrafiltration, QT was mostly associated to the lipid phase, confirming its affinity for the vesicle bilayer, especially in the case of T-ETO_2.7_-QT, as expected, due to the higher PC concentration, leading to the formation of multivesicular vesicles. The aliquot of QT in the water phase is in agreement with the findings about localization of the molecule within the bilayer at the polar–nonpolar interface (Appendix A), forming reversible physico-chemical complexes with PC [53]. This QT position could alter the packing density of the bilayers, softening the membrane, finally affecting the EC, as previously found for mangiferin-loaded ETO and T-ETO [41,52]. Nonetheless, the highest EC obtained in the case of T-ETO_2.7_-QT suggests that the presence of T80 can better stabilize the vesicles based on PC 2.7%, possibly improving QT interaction with PC, thus maintaining the drug associated to the bilayer supramolecular structure, as tentatively depicted in Appendix A.

Some authors demonstrated that the flavonoids antioxidant activity depends not only on their structural characteristics, but also on their location in PC membrane systems [54]. ACL values suggest that QT encapsulation within T-ETO based on the highest PC amount could improve QT antioxidant capacity. Thus, the higher amount of PC in T-ETO_2.7_-QT could account for a stronger ability to maintain QT associated to the vesicle interface, in agreement with EC results. Accordingly, stability results confirm this hypothesis, suggesting that both T80 presence and PC 2.7% could contribute to improve the QT interaction within the PC bilayer of the multivesicular vesicles, conversely in the case of ETO_0.9_-QT, the drug could be more exposed to oxidative degradation, in agreement with antioxidant activity results [54].

For IVRT, different types of membranes were considered, i.e., the hydrophilic NY, and the hydrophobic PTFE (Appendix A) with the aim to investigate the membrane effect on QT release. Indeed, despite synthetic membranes are recommended by the FDA, being commercially available, chemically inert, and typically employed for quality control purposes, the extent of drug release can be influenced by some factors, such as drug physicochemical properties, membrane chemical character, pore size and thickness, as well as the compatibility of the membrane with the donor and the receptor components [27,28,29,30,55]. IVRT data suggest that NY could restrain QT release, as previously found in the case of ketoprofen [30]. The different NY and PTFE behavior could be attributed to (i) a partial vesicle association within the membrane structure, and to (ii) an incompatibility between the hydrophilic character of NY and the more lipophilic character of the donor and receiving phases. The QT release discrepancies evidenced the crucial importance of the membrane choice, as previously demonstrated in a study about a gallic acid-loaded gel [56].

With regard to IVPT, some consideration should be carried out. The higher *p* value obtained in the case of T-ETO_0.9_-QT with respect to ETO_0.9_-QT, suggests that the presence of T80 can induce an alteration of packing density in PC bilayers, resulting in a QT preferential distribution towards the STRAT-M^®^ membrane [41]. On the other hand, the lowest *p* value calculated in the case of T-ETO_2.7_-QT indicates a preferential distribution of QT for the vehicle with respect to the membrane [35]. Therefore, the vesicles based on the highest PC concentration can retain QT more tightly, thus controlling its permeation kinetic, as confirmed by the longest Tlag and the lowest Kp values obtained in the case of T-ETO_2.7_-QT (Appendix A). Conversely, the highest Kp and D values achieved by T-ETO_0.9_-QT, confirms the capability of T80 associated to PC vesicles to enhance QT diffusion through the membrane, resulting in the highest amount of QT diffused in 24 h (A_QT_ total).

This behavior was confirmed also using SCE, suggesting the capability of PC vesicles to promote QT permeation through the *stratum corneum*. A_QT_ values reflect the trend, indicating that T-ETO can improve QT diffusion with respect to ETO_0.9_-QT.

Notably, IVPT parameters obtained by SCE were not compared to those obtained by STRAT-M^®^, since the thickness of SCE was 14-fold lower with respect to STRAT-M^®^ (Appendix A). In addition, *p* and D values were not considered, being directly related to the membrane thickness, thus dramatically higher with respect to those obtained by the polymeric membrane.

Concerning biological data, the role of ROS in would healing has been well studied, indeed an over production of ROS inhibits wound healing, while a homeostatic concentration can improve this process [57], since ROS can play a second messenger role. It is possible that the slow release of QT, loaded in ETO and T-ETO, could prevent the formation of ROS (either directly or via the activation of NRF2), leading to a slower wound closure response. In addition, considering that NFkB is involved in would healing, NRF2 activation exploited by QT could inhibit the activation of NFkB, delaying cellular proliferation and recovery from the scratch wound [58].

The more efficient ability of T-ETO_2.7_-QT in inhibiting cell migration for both HaCaT and HT-144 cells is in accordance with their superior antioxidant activity demonstrated by PCL test. At this regard, the possible involvement of QT antioxidant activity in migration properties will be further investigated.

## 5. Conclusions

This study enabled to design ETO and T-ETO suitable for QT loading and topical delivery, by a simple, cost effective and easily scalable method. The vesicle composition strongly affected size distribution, loading capacity and stability of vesicles. Particularly, the highest PC concentration and the presence of T80 in T-ETO_2.7_-QT on one hand enlarged the vesicle size with respect to ETO_0.9_-QT, on the other these parameters contributed to improve QT entrapment capacity, antioxidant activity and stability. Notably, T-ETO_0.9_-QT and T-ETO_2.7_-QT promoted QT permeation with respect to ETO_0.9_-QT, as demonstrated by Franz cells associated to Strat-M^®^ and to human SCE membranes. Nevertheless, further study will be requested in order to assess the distribution of QT in the different human skin layers. Moreover, the encouraging biological results here presented suggest the possibility to employ T-ETO_2.7_-QT as adjuvant in the treatment of melanoma, or possibly in other skin iperproliferative conditions, including psoriasis. Remarkably, the possibility to administer QT by T-ETO can represent an innovative approach to hamper the toxicity issues related to dietary intake, avoiding the gastrointestinal system absorption. The preliminary data shown in this work strongly support further preclinical studies aimed to understand the potential applicability of T-ETO in anticancer topical therapies.

## Figures and Tables

**Figure 1 pharmaceutics-14-01038-f001:**
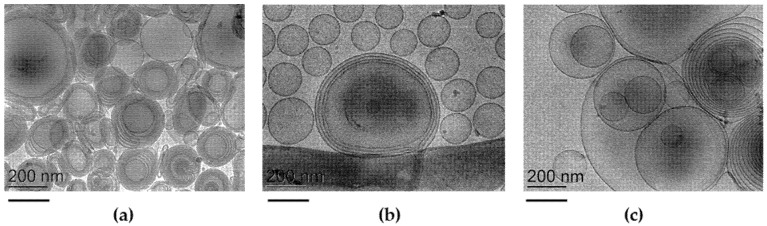
Cryo-transmission electron microscopy images (Cryo-TEM) of ETO_0.9_-QT (**a**), T-ETO_0.9_-QT. (**b**) and T-ETO_2.7_-QT (**c**). The bars correspond to 200 nm.

**Figure 2 pharmaceutics-14-01038-f002:**
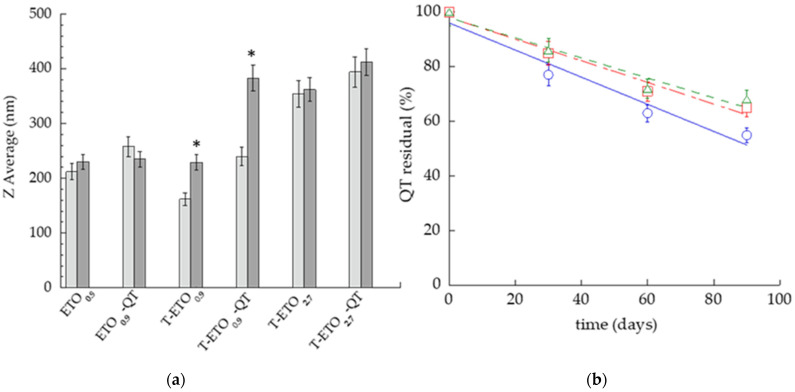
Effect of storage on ethosomes and transethosomes kept at 22 °C for 3 months. (**a**) Z Average mean diameters measured by PCS 1 (light grey column) and 90 (grey column) days after preparation, * *p* values < 0.05; (**b**) QT residual content in ETO_0.9_-QT (blue circle); T-ETO0.9-QT (red square) and T-ETO_2.7_-QT (green triangle), the percentage refers to the total QT content evaluated after sample preparation.

**Figure 3 pharmaceutics-14-01038-f003:**
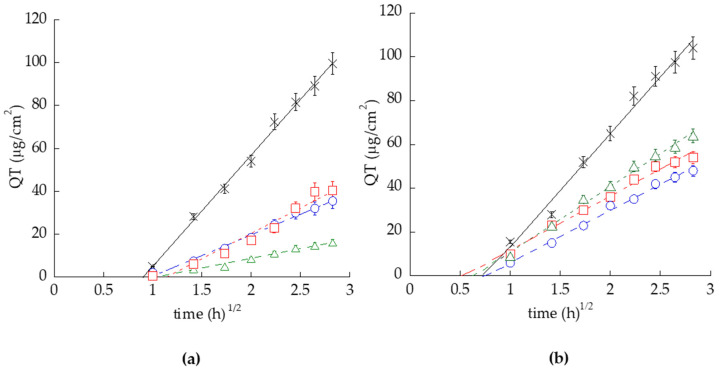
QT release kinetics from ETO_0.9_-QT (blue circles), T-ETO_0.9_-QT (red squares), T-ETO_2.7_-QT (green triangles), and SOL-QT (black crosses), as determined by Franz cell associated to NY (**a**) and PTFE (**b**). Data are the mean of 6 independent experiments ± s.d.

**Figure 4 pharmaceutics-14-01038-f004:**
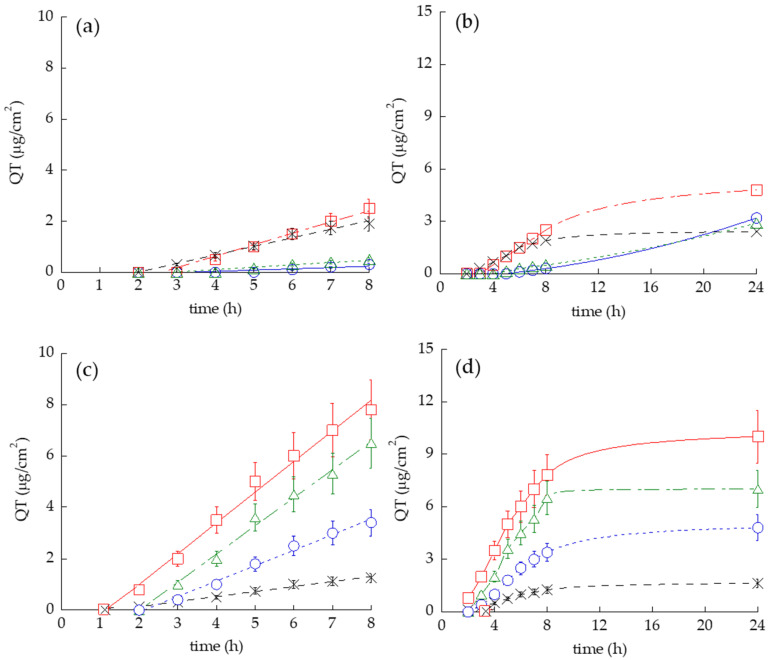
QT permeability kinetics from ETO_0.9_-QT (blue circles), T-ETO_0.9_-QT (red squares), T-ETO_2.7_-QT (green triangles), and SOL-QT (black crosses), as determined by Franz cell associated to Strat-M^®^ (**a**,**b**) and SCE (**c**,**d**). Data are the mean of 12 independent experiments ± s.d.

**Figure 5 pharmaceutics-14-01038-f005:**
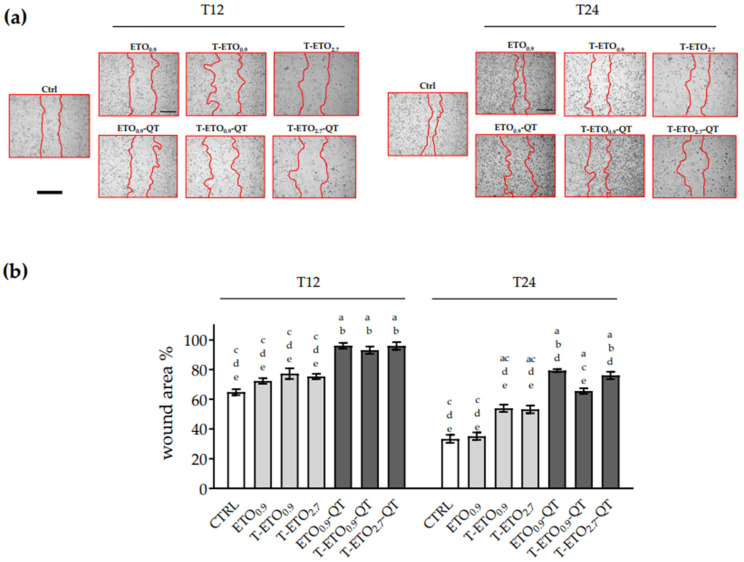
Effect of QT-loaded or unloaded nano-vesicular forms on the wound closure in HaCaT cells. (**a**) Scratch was performed on confluent monolayer of HaCaT cells, and different images were taken to measure wound area at 12 h (**left panel**) and 24 h (**right panel**) time points (scale bar 400 μm). (**b**) Quantification of the wound area at each time point via ImageJ. Data are shown as percent of 0 h. Data are the results of 3 independent experiments performed in triplicate. a: *p* < 0.05 vs. Ctrl; b: *p* < 0.05 vs. QT-unloaded formulations; c: *p* < 0.05 vs. ETO_0.9_-QT; d: *p* < 0.05 vs. T-ETO_0.9_-QT; e: *p*< 0.05 vs. T-ETO_2.7_-QT by two-way ANOVA.

**Figure 6 pharmaceutics-14-01038-f006:**
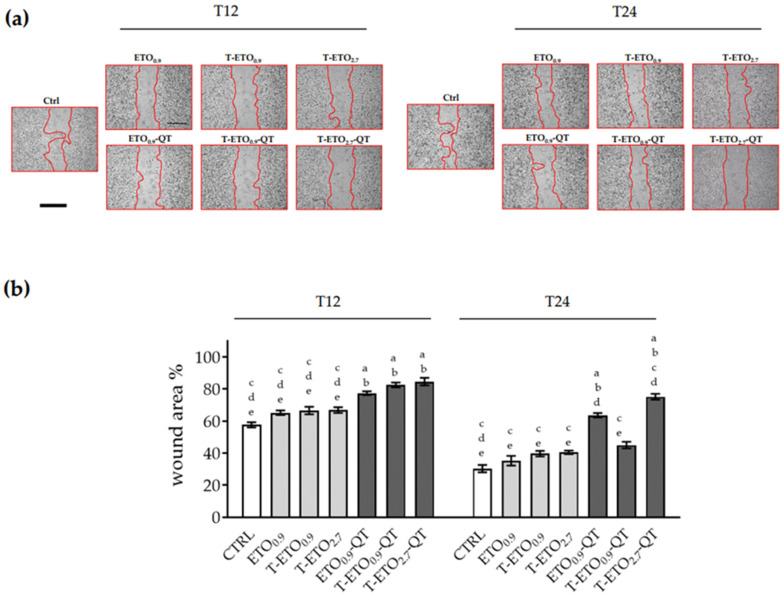
Effect of QT-loaded or unloaded nano-vesicular forms on the wound closure in HT-144 melanoma cells. (**a**) Scratch was performed on confluent monolayer of HT-144 cells, and different images were taken to measure wound area at 12 (**left panel**) and 24 h (**right panel**) time points (scale bar 400 μm). (**b**) Quantification of the wound area at each time point via ImageJ. Data are shown as percent of 0 h. Data are the results of three independent experiments performed in triplicate. a: *p* < 0.05 vs. Ctrl; b: *p* < 0.05 vs. QT-unloaded formulations; c: *p* < 0.05 vs. ETO_0.9_-QT; d: *p* < 0.05 vs. T-ETO_0.9_-QT; e: *p* < 0.05 vs. T-ETO_2.7_-QT by two-way ANOVA.

**Figure 7 pharmaceutics-14-01038-f007:**
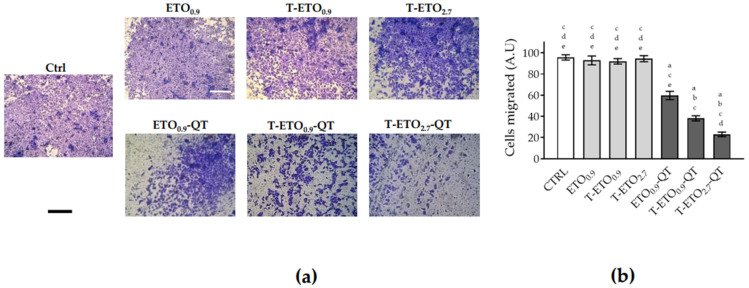
HaCaT cell migration after treatment with QT-loaded or unloaded nano-vesicular forms. (**a**): representative images of HaCat cell migration after 24 h of pretreatment with QT loaded or unloaded formulations (scale bar 200 μm). After 24 h of pretreatment, HaCaT cells were detached, seeded in 8 μm pore size transwells and incubated for 24 h. (**b**): ImageJ quantification of the migrated cells. Data are shown as average of 5 picture fields (20× magnification). a: *p* < 0.05 vs. Ctrl; b: *p* < 0.05 vs. QT-unloaded forms; c: *p* < 0.05 vs. ETO_0.9_-QT; d: *p* < 0.05 vs. T-ETO_0.9_-QT; e: *p* < 0.05 vs. T-ETO_2.7_-QT by two-way ANOVA.

**Figure 8 pharmaceutics-14-01038-f008:**
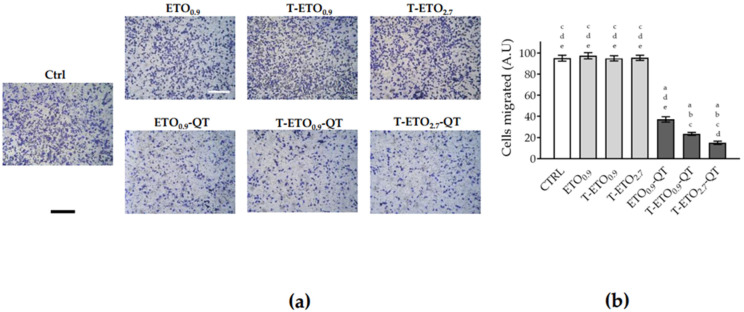
HT-144 melanoma cells migration after treatment with QT-loaded or unloaded nano-vesicular forms. (**a**): representative images of HT-144 cell migration during 24 h treatment with QT loaded or unloaded formulations (scale bar 200 μm). HT-144 cells were detached, seeded in 8 μm pore size transwells and incubated for 24 h with the nano-vesicular forms. (**b**): ImageJ quantification of the migrated cells. Data are shown as average of 5 picture fields (20× magnification). a: *p* < 0.05 vs. Ctrl; b: *p* < 0.05 vs. QT-unloaded formulations; c: *p* < 0.05 vs. ETO_0.9_-QT; d: *p* < 0.05 vs. T-ETO_0.9_-QT; e: *p* < 0.05 vs. T-ETO_2.7_-QT by two-way ANOVA.

**Table 1 pharmaceutics-14-01038-t001:** Composition of the indicated formulations.

Formulation Code	PC ^1^ % *w*/*w*	Ethanol % *w*/*w*	T80 ^2^ % *w*/*w*	Water % *w*/*w*	QT ^3^ % *w*/*w*
ETO_0.9_	0.9	29.10	-	70.0	-
ETO_0.9_-QT	0.9	29.05	-	70.0	0.05
T-ETO_0.9_	0.9	29.10	0.3	69.7	-
T-ETO_0.9_-QT	0.9	29.05	0.3	69.7	0.05
T-ETO_2.7_	2.7	27.00	0.3	69.7	-
T-ETO_2.7_-QT	2.7	26.95	0.3	69.7	0.05

^1^: soy phosphatidylcholine; ^2^: polysorbate 80; ^3^: quercetin.

**Table 2 pharmaceutics-14-01038-t002:** Size distribution parameters of ethosomes and transethosomes, as determined by PCS.

Formulation Code	Time (Days)	Z Average (nm) ± s.d.	Typical Intensity Distribution		Dispersity Index ± s.d.
			nm *	Area (%) *	
ETO_0.9_	1	212.25 ± 14.20	227.4	100	0.12 ± 0.01
	90	230.52 ± 25.51	245.5	100	0.14 ± 0.03
ETO_0.9_-QT	1	258.18 ± 21.42	230.4	96	0.24 ± 0.02
	90	235.45 ± 20.20	220.7	97	0.22 ± 0.03
T-ETO_0.9_	1	161.90 ± 11.32	174.5	100	0.13 ± 0.02
	90	229.15 ± 42.42	201.3	100	0.15 ± 0.03
T-ETO_0.9_-QT	1	240.12 ± 21.30	230.4	91	0.24 ± 0.01
	90	383.25 ± 47.61	290.2	87	0.26 ± 0.03
T-ETO_2.7_	1	354.00 ± 35.53	322.6	92	0.25 ± 0.02
	90	362.00 ± 24.32	330.5	90	0.24 ± 0.01
T-ETO_2.7_-QT	1	394.00 ± 32.21	350.2	91	0.23 ± 0.02
	90	412.21 ± 27.40	320.8	88	0.26 ± 0.01

s.d.: standard deviation; *: main peak; data are the mean of 3 independent determinations on different batches.

**Table 3 pharmaceutics-14-01038-t003:** Entrapment capacity and antioxidant activity of the indicated forms.

Formulation Code	EC ^1^ (%)	ACL ^2^ (μmol TE/g)
ETO_0.9_-QT	56.44 ± 1.00	3.26 ± 0.17
T-ETO_0.9_-QT	59.19 ± 1.04	3.16 ± 0.10
T-ETO_2.7_-QT	64.10 ± 0.55	5.06 ± 0.25
SOL-QT	-	3.82 ± 0.05

^1^: Entrapment capacity, as defined in Equation (1); ^2^: Antioxidant Capacity of Lipid soluble compounds; data are the mean of 6 independent experiments ± s.d.

**Table 4 pharmaceutics-14-01038-t004:** IVRT parameters of the indicated formulations.

Formulation Code	R_QT_ ^1^ ± s.d. (μg/cm^2^/h)	T_lag_ ^2^± s.d. (h)	A_QT_ ^3^ ± s.d. (μg/cm^2^)	M_QT_ ^4^ ± s.d. (μg)
ETO_0.9_-QT	18.84 ± 0.91 ^N^	1.00 ± 0.10 ^N^	35.4 ± 1.2 ^N^	18.0 ± 1.1 ^N^
	23.73 ± 0.22 ^P^	0.70 ± 0.00 ^P^	48.0 ± 2.4 ^P^	1.2 ± 0.1 ^P^
T-ETO_0.9_-QT	24.63 ± 0.93 ^N^	1.04 ± 0.12 ^N^	40.4 ± 2.0 ^N^	14.0 ± 1.2 ^N^
	24.68 ± 0.92 ^P^	0.50 ± 0.00 ^P^	54.0 ± 2.2 ^P^	3.0 ± 0.5 ^P^
T-ETO_2.7_-QT	8.99 ± 0.91 ^N^	1.05 ± 0.33 ^N^	16.1 ± 0.4 ^N^	22.0 ± 2.2 ^N^
	30.00 ± 1.20 ^P^	0.62 ± 0.00 ^P^	64.0 ± 2.4 ^P^	1.2 ± 0.1 ^P^
SOL-QT	49.13 ± 2.52 ^N^	0.90 ± 0.01 ^N^	99.6 ± 3.4 ^N^	34.0 ± 2.0 ^N^
	51.77 ± 1.91 ^N^	0.56 ± 0.02 ^P^	104.0 ± 2.8 ^P^	9.0 ± 0.1 ^P^

^1^: QT release rate; ^2^: lag-time; ^3^: amount of QT released after 8 h; ^4^: M_QT:_ amount of QT associated to the membrane after 8 h; _N_: as determined by NY membrane; _P_: as determined by PTFE membrane; QT was always 0.5 mg/mL; data are the mean of 6 independent Franz cell experiments ± s.d.

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
