# Peer review of "Ethosomes and Transethosomes as Cutaneous Delivery Systems for Quercetin: A Preliminary Study on Melanoma Cells"

_pharmaceutics, 2022, doi:10.3390/pharmaceutics14051038_

Round 1

Reviewer 1 Report

The authors report the development, characterization, and in vitro evaluation of ethosomes and transethosomes containing quercetin. The methodology is detailed and well stablished. Furthermore, it answers the questions raised by the authors. However, the authors should address the following comments before publication: 

1)      Introduction: The authors are advised to adjust the information about membrane and their choice (starting at line 74). This information is of particular interest in the discussion, to allow the flow of the text. 

2)      At the topic “2.10.2 Wound Healing assay”, did the authors use serum-free medium? Please provide the information.

3)      Correct the numbering of  ”Statistical analysis” topic (2.11)?

4)      English language mistakes may impair the comprehension or cause erroneous understanding of the sentences. Please review/rewrite sentences with possible mistakes, such as:

  • Topic 3.1, lines 344-347: “Since research studies about phospholipid vesicles have demonstrated a correlation between phospholipid concentration and drug entrapment efficiency [16], in T-ETO the PC concentration was 3-fold increased, reaching 2.7% w/w.’’. 
  • Lines 686-87: “This effect resulted even more evident after 24h…” 

5)      In topic “3.3.1. Morphology and size” it would be nice to statistically compare the Z average between 1 and 90 days of each treatment. Please, indicate it in the text if the test was already performed.

6)   Please, point out that the results of the EC are in Table 4, in topic “3.3.1. QT entrapment capacity (EC)” . Furthermore, correct the topic number (3.3.2). 

7) Did the authors tested empty vesicles in topic “3.4. Evaluation of QT antioxidant activity”? This should be tested to affirm that the antioxidant activity obtained is exclusively due to quercetin. Empty vesicles may also have some activity as observed in cell viability and wound healing assays. Please, also highlight that the viability wasn’t higher than 80% after treatments in both cell lines. 

8)      Figures 3, 4 and S2 should indicate statistical significance to confirm the difference between SOL-QT and the other treatments. 

Author Response

Reviewer 1

The authors report the development, characterization, and in vitro evaluation of ethosomes and transethosomes containing quercetin. The methodology is detailed and well stablished. Furthermore, it answers the questions raised by the authors. However, the authors should address the following comments before publication: 

Point 1)      Introduction: The authors are advised to adjust the information about membrane and their choice (starting at line 74). This information is of particular interest in the discussion, to allow the flow of the text. 

Answer 1)    We thank the reviewer for this suggestion, the introduction has been revised accordingly. “ …. In the case of IVRT, usually synthetic membranes based on nylon, mixed cellulose esters, or polytetrafluoroethylene are used. With regard to IVPT, despite natural membranes can achieve predictable results regarding in vivo drug permeation, many factors restrict their employment, such as variability of results, scarce availability, and high costs [27–31]. To overcome these drawbacks, recently peculiar synthetic membrane systems, able to reproduce the peculiar stratum corneum assembly, have been proposed [31]. Particularly, the multiple polymeric membrane system named Strat-M®, based on two polyether sulfone layers overlapped to one polyolefin bottom layer, has skin-like tortuous porous structure [32]. Moreover, the impregnation with synthetic lipids further confers to Strat-M® a resemblance with the skin, characterized by hydrophilic and lipophilic compartments, providing barrier properties [32]. Therefore, in this study the diffusion performances of QT loaded-ETO and T-ETO were evaluated by Franz cell, using Nylon (NY) and polytetrafluoroethylene (PTFE) membranes for IVRT, while, for IVPT, both STRAT-M® and human stratum corneum-epidermis (SCE) have been employed.….”

Point 2)      At the topic “2.10.2 Wound Healing assay”, did the authors use serum-free medium? Please provide the information.

Answer 2) For the “Wound Healing assay”, we used complete medium. We apologize for the missing information.

Point 3)      Correct the numbering of  ”Statistical analysis” topic (2.11)?

Answer 3)      The numbering has been corrected accordingly.

Point 4)      English language mistakes may impair the comprehension or cause erroneous understanding of the sentences. Please review/rewrite sentences with possible mistakes, such as:

  • Topic 3.1, lines 344-347: “Since research studies about phospholipid vesicles have demonstrated a correlation between phospholipid concentration and drug entrapment efficiency [16], in T-ETO the PC concentration was 3-fold increased, reaching 2.7% w/w.’’. 
  • Lines 686-87: “This effect resulted even more evident after 24h…” 

Answer 4)    We thank the reviewer for this observation. The phrases have been rewritten: Since research studies have demonstrated that drug entrapment efficiency is influenced by the phospholipid concentration of vesicles [17], the PC concentration of T-ETO has been increased, up to 2.7% w/w (3-fold). ….

“Conversely, after 24 h of pretreatment with ETO0.9-QT, T-ETO0.9-QT and T-ETO2.7-QT, a significant and dramatic reduction of cellular migration was observed.”

Point 5)      In topic “3.3.1. Morphology and size” it would be nice to statistically compare the Z average between 1 and 90 days of each treatment. Please, indicate it in the text if the test was already performed.

Answer 5)      The Z average between 1 and 90 days of each treatment was statistically compared in the original version of the manuscript, as reported in section 2.5 of the method section, as well as indicated with asterisks in Figure 2 and reported in the legend “*p values < 0.05”. As suggested by the reviewer, the phrase “The increase of vesicle size was statistically significant both for T-ETO and T-ETO0.9-QT” has been added in the text at lines 432-433.

Point 6)   Please, point out that the results of the EC are in Table 4, in topic “3.3.1. QT entrapment capacity (EC)” . Furthermore, correct the topic number (3.3.2). 

Answer 6)     At line 399 the phrase “Table 3 reports the EC results.” has been inserted. Indeed in the revised version Table 1 has been moved to Supplementary data section, therefore Table 4 is now Table 3. The topic number has been corrected accordingly.

Point 7) Did the authors tested empty vesicles in topic “3.4. Evaluation of QT antioxidant activity”? This should be tested to affirm that the antioxidant activity obtained is exclusively due to quercetin. Empty vesicles may also have some activity as observed in cell viability and wound healing assays. Please, also highlight that the viability wasn’t higher than 80% after treatments in both cell lines. 

Answer 7) We thank the reviewer for the observation. The empty vesicles were tested at the same time as those with the active and, under the same experimental conditions, did not give any activity to the PCL assay (as indicated at  lines 411-412).

Concerning the viability (Figure S2), we would like to underline that we did not test the cytotoxicity of SOL-QT on HaCaT and HT-144 cell lines; indeed, the aim of the biological activity studies was to compare the efficiency of the different formulation (ETO0.9, T-ETO0.9, T-ETO2.7 unloaded or loaded with QT) in decrease proliferation, migration, and invasion of these two cell lines. As regard to the statistical analysis of the cytotoxicity of QT-loaded and unloaded forms (Figure S3), the aim of MTT test performed was to evaluate the highest non-cytotoxic concentration able to not induce a decrease in cell viability higher than 35%. Therefore, by two-way ANOVA, followed by Tukey Test as post hoc test, we selected 2.5 μg/mL as the proper dose to be used. Anyway, according to this explanation, we changed Figure S2, its relative caption and the 3.7.1 paragraph as follows: “In both cell lines a significant decrease in cell viability was detected only for 5 μg/mL of QT (≈ 50%), compared to untreated cells (Figure S2). Based on these results, we selected 2.5 μg/mL of QT-loaded and unloaded formulations as the highest non-cytotoxic concentration for the subsequent assays, since it did not induce a decrease in cell viability higher than 35%.”

Point 8)      Figures 3, 4 and S2 should indicate statistical significance to confirm the difference between SOL-QT and the other treatments. 

Answer 8) We thank the reviewer for the suggestion and we have now indicated the statistical significance as requested. In the case of IVRT (Figure 3) and IVPT (Figure 4), to evaluate the difference between SOL-QT and the vehicles, the statistical analysis was performed comparing RQT in the case of IVRT and KP in the case of IVPT. Comments have been inserted at lines 454, 517-518.

With regard to Figure S2, please see the previous answer.

Reviewer 2 Report

The manuscript Ethosomes and transethosomes as cutaneous delivery systems 2 for quercetin: a preliminary study on melanoma cells developed a new formulation of quercetin drug delivery. The paper showed physicochemical and biological results.

Please, answer some questions

  1. Why, did you choice ethosone and transethosome? Which difference between?
  2. The antioxidant activity not showed difference using nanotechnology. Please, show statistical and.. if is necessary to use formulation? What benefits to justify the use.
  3. I think the paper is so long, if possible summarize.

Author Response

Reviewer 2

The manuscript Ethosomes and transethosomes as cutaneous delivery systems 2 for quercetin: a preliminary study on melanoma cells developed a new formulation of quercetin drug delivery. The paper showed physicochemical and biological results.

Please, answer some questions

Point 1) Why, did you choice ethosone and transethosome? Which difference between?

Answer 1) As reported in the revised introduction, ETO are phospholipid vesicular systems usually constituted of phosphatidylcholine (PC), water and ethanol (20-45 % v/v) [16]. These vesicular systems can be defined as the second generation of liposomes, being characterized by higher malleability, longer stability and better capability to entrap insoluble molecules [17]. Indeed, in ETO, the presence of PC and ethanol exerts a penetration enhancement effect, due to a disorganization of the stratum corneum barrier, leading to an improvement of drug permeability through the skin [18,19]. Moreover, in order to confer to ETO vesicles an even more flexible structure, the so-called transethosomes (T-ETO) can be formulated by the addition of non-ionic surfactants, as well as of polyethylene glycol, to PC ethanol solution [20–22]. In T-ETO the presence of surfactants, employed as “edge activators”, should further improve the vesicle transdermal potential, making them a novel and promising technology for the treatment of different cutaneous conditions [23–25].

Point 2) The antioxidant activity did not showed difference using nanotechnology. Please, show statistical and.. if is necessary to use formulation? What benefits to justify the use.

Answer 2) We have detected some antioxidant properties using nanotechnology, as reported in the revised text: “Conversely, QT-loaded vesicles showed an antioxidant activity, remarkably improved in the case of T-ETO2.7-QT. Indeed, ETO0.9-QT and T-ETO0.9-QT possess a similar antioxidant capacity (differences not statistically significant), lower with respect to QT ethanolic solution (SOL-QT), while T-ETO2.7-QT displayed the highest antioxidant capacity, being 1.32-fold higher with respect to SOL-QT and 1.6-fold higher with respect to T-ETO0.9-QT (ACL values extremely significantly different; p < 0.0001).”

In the revised manuscript the statistical evaluation has been added.

Moreover, we would like to underline that one of the advantages of using ETO and T-ETO formulations as loading systems for QT is related to their biocompatibility and suitability for cutaneous application. On the contrary, SOL-QT could result irritant to the applied cutaneous tissue.

Point 3) I think the paper is so long, if possible summarize.

Answer 3) Dear Reviewer, we do agree with you about the length of the manuscript, but this is the consequences of having acknowledged all the requests of the previous round of reviewers (n 4 first round and n 5 second round) asking several more experiments as it can be checked in the chronology of the manuscript dashboard (2 revisions before this re-submission). To follow your request and to shorten the manuscript, Table 1, 6 and 7 have been moved to supplementary data (Tables S1, S3 and S4 respectively), moreover the text has been modified moving the discussion in a separate section.

Reviewer 3 Report

The manuscript by Ferrara et al. “Ethosomes and transethosomes as cutaneous delivery systems for quercetin: a preliminary study on melanoma cells” seems interesting and can serve as a platform for topical drug delivery. The overall study design is good. Thus, I recommend publication.

Author Response

Reviewer 3

The manuscript by Ferrara et al. “Ethosomes and transethosomes as cutaneous delivery systems for quercetin: a preliminary study on melanoma cells” seems interesting and can serve as a platform for topical drug delivery. The overall study design is good. Thus, I recommend publication.

We thank the reviewer for her/his kind comment.

Reviewer 4 Report

this study on ethosome and transethosome encapsulation of quercitin appears to have been carried out with a thorough approach and has produced some promising results. the methodology is solid and the flow of information in the results and discussion is presented nicely. there are only a few minor concerns:

-a rewrite of the introduction should be considered. the current introduction is a bit unfocused and at the absolute minimum would benefit from being broken up into shorter sections for readability.

-the current introduction also does not accomplish the goals of highlighting the novelty of this work and framing its contribution to the advancement of the field very clearly. it states that specific properties make ETO suitable for the treatment of melanoma, but that appears to be the topic of this study. has this already been done? where does this study fit in the context of the current state of the field then?

-the scale bar in figure 2b is not readable. consider change the color of all 3 scale bars in that figure for consistency and readability

-the plots in figure 4 could be increased in size without any bad effects on the formatting, it seems. this would help in resolving the different markers in the a and b subplots.

-in several tables the average is presented with more significant digits than the standard deviation. if there is a good reason for this, maybe the authors could mention it in the caption?

other than these relatively minor issues the work as a whole appears to be of very high quality and the presentation in general looks polished. 

Author Response

Reviewer 4

this study on ethosome and transethosome encapsulation of quercitin appears to have been carried out with a thorough approach and has produced some promising results. the methodology is solid and the flow of information in the results and discussion is presented nicely. there are only a few minor concerns:

Point 1) a rewrite of the introduction should be considered. the current introduction is a bit unfocused and at the absolute minimum would benefit from being broken up into shorter sections for readability.

Answer 1) We thank the reviewer for this suggestion. The introduction has been revised accordingly.

Point 2) the current introduction also does not accomplish the goals of highlighting the novelty of this work and framing its contribution to the advancement of the field very clearly. it states that specific properties make ETO suitable for the treatment of melanoma, but that appears to be the topic of this study. has this already been done? where does this study fit in the context of the current state of the field then?

Answer 2) As reported in the introduction, that has been changed, “……In T-ETO the presence of surfactants, employed as “edge activators”, should further improve the vesicle transdermal potential, making them promising in the treatment of different cutaneous diseases, including melanoma [23–25]. Indeed ETO and T-ETO can enhance the transdermal delivery of the entrapped drugs, since the vesicles can pass through the stratum corneum barrier, promoting the drug passage towards the dermis [19,20]. Nonetheless, the possibility to treat melanoma by topical application of QT loaded ETO and T-ETO has never been explored. Thus, in this study firstly ETO and T-ETO were designed and compared for QT loading, then their transdermal potential and suitability as possible candidates in the adjuvant therapy of melanoma have been investigated.”

Point 3) the scale bar in figure 2b is not readable. consider change the color of all 3 scale bars in that figure for consistency and readability

Answer 3) The scale bars of Figure 3 (a, b and c) have been depicted out of the panel boxes for better clarity

Point 4) the plots in figure 4 could be increased in size without any bad effects on the formatting, it seems. this would help in resolving the different markers in the a and b subplots.

Answer 4) We thank the reviewer for this suggestion. The plots of Figure 4 have been modified as requested.

Point 5) in several tables the average is presented with more significant digits than the standard deviation. if there is a good reason for this, maybe the authors could mention it in the caption?

Answer 5) We thank the reviewer for this observation, the digits of the standard deviation in Tables 2-4, S3 and S4  have been corrected accordingly.

other than these relatively minor issues the work as a whole appears to be of very high quality and the presentation in general looks polished. 

Reviewer 5 Report

Authors in the manuscript have conducted a study to evaluate ethosomes and transethosomes as a delivery vehicles for cutaneous delivery of quercetin. After critical review, the manuscript can be accepted for publication after addressing the comments below.

Lines 29-30: Please change franz cell tests to permeation studies conducted with strat m membrane revealed that………….

Line 46: Change At this regard, to “In this regard”. Please perform an editorial check before submission of next version. Please reword underlined in line 392 to it should be noted that……

Lines 385 to 387: Nice explanation, if possible please provide a graphical representation of this (not mandatory), but would be very interesting

Line 202: Please change to QT amount released (ug/cm2) was plotted

Section 2.7.2: Please change the name to In vitro permeation test, since skin was not used in the study. You can add IVPT using Strat-M membrane. Also, please provide the details of the experimentation in this section rather than discussing the parameters. Methods such as sampling time, what kind of franz cell was used, what was the sample volume and interval and such should be covered in the methods section.

HPLC: Please provide the retention time and sample volume,

Global comment on results and discussion: Authors have put so much effort to justify the results, however, the organization of the results is confusing. It is advised to first present the results of the experiments, followed by the reasoning and discussion of findings.

It is recommended to cut down unnecessary information from the results section.  Also, please provide advantages of using Strat M as a permeation membrane instead of skin in the results and discussion section.

Conclusions: Please remove preformulative from the sentence as it was not a preformuation study.

Author Response

Reviewer 5

Authors in the manuscript have conducted a study to evaluate ethosomes and transethosomes as a delivery vehicles for cutaneous delivery of quercetin. After critical review, the manuscript can be accepted for publication after addressing the comments below.

Point 1) Lines 29-30: Please change franz cell tests to permeation studies conducted with strat m membrane revealed that………….

Answer 1) We thank the reviewer for this suggestion. We would like to underline that permeation studies were conducted both by STRAT-M and stratum corneum-epidermis (SCE). Accordingly, we changed the abstract with the phrase “In vitro permeation studies revealed…” at lines 29-30.

Point 2) Line 46: Change At this regard, to “In this regard”. Please perform an editorial check before submission of next version. Please reword underlined in line 392 to it should be noted that……

Answer 2) The terms have been changed accordingly.

Point 3) Lines 385 to 387: Nice explanation, if possible please provide a graphical representation of this (not mandatory), but would be very interesting.

Answer 3) We thank the reviewer. A graphical representation of ETO in the presence and in the absence of QT has been reported in Figure S3. Moreover a graphical representation of T-ETO2.7  has been reported in Figure S4. 

Point 4) Line 202: Please change to QT amount released (ug/cm2) was plotted.

Answer 4) The term “released” has been inserted accordingly.

Point 5) Section 2.7.2: Please change the name to In vitro permeation test, since skin was not used in the study. You can add IVPT using Strat-M membrane. Also, please provide the details of the experimentation in this section rather than discussing the parameters. Methods such as sampling time, what kind of franz cell was used, what was the sample volume and interval and such should be covered in the methods section.

Answer 5) We thank the reviewer, nonetheless we would like to underline that, as reported in section 2.7 and Table S1, IVPT tests were performed both using STRAT-M® and human stratum corneum-epidermis (SCE). The details of the experimentation, as well as methods, were yet provided in section 2.7, such as the kind of Franz cell used (Franz vertical cells (orifice diameter 0.9 cm; PermeGear Inc. Hellertown, PA, USA), the sample volume (0.5 mL) and interval (comprised between 1 and 24 h )(line….). In section 2.7.1 and 2.7.2 is presented the way of processing data and the method of calculation of parameters. We prefer to maintain here this part, avoiding to lengthen the result section.

Point 6) HPLC: Please provide the retention time and sample volume,

Answer 6) the retention time and sample volume have been specified accordingly: “Injection volume and retention time were 5 ml and 2.8 min respectively “ at lines 238-239.

Point 7) Global comment on results and discussion: Authors have put so much effort to justify the results, however, the organization of the results is confusing. It is advised to first present the results of the experiments, followed by the reasoning and discussion of findings.

Answer 7) We thank the reviewer, we modified the results and implemented the discussion section accordingly.

Point 8) It is recommended to cut down unnecessary information from the results section.  Also, please provide advantages of using Strat M as a permeation membrane instead of skin in the results and discussion section.

Answer 8) Dear Reviewer, we do agree with you about the length of the manuscript, but this is the consequences of having acknowledged all the requests of the previous round of reviewers (n 4 first round and n 5 second round) asking several more experiments as it can be checked in the chronology of the manuscript dashboard (2 revisions before this re-submission). Anyway, to shorten the manuscript, Table 1, 6 and 7 have been moved to supplementary data (Tables S1, S3 and S4 respectively) and some paragraphs have been removed.

As reported above, as well as in the manuscript, IVPT tests were performed using both STRAT-M® and skin (human stratum corneum-epidermis, SCE). Anyway, the advantages of using the different natural or synthetic membranes are reported in the introduction at lines 82-86, we did not rewrite them also in the result and discussion section to avoid lengthening the text.

Point 9) Conclusions: Please remove preformulative from the sentence as it was not a preformuation study.

Answer 9) The term preformulative has been removed accordingly

This manuscript is a resubmission of an earlier submission. The following is a list of the peer review reports and author responses from that submission.

Round 1

Reviewer 1 Report

Overall, Efficacy verification of ethosome using in vitro Franz cells is not sufficient. It is necessary to perform the Franz cell experiment using the ex-vivo skin, which has the stratum corneum rather than the membranes.

1. The delivered amount (and % to total amount) in SC, dermis, and receiver regarding time should be shown.
2. The receptor solution is 50% ethanol, which itself is a penetration enhancer, and why 50% ethanol was used for the receptor.
3. Please add the explanation about the delivery mechanism of nano-particles with region of  200-400 nm in average size through the stratum corneum .

Author Response

Overall, Efficacy verification of ethosome using in vitro Franz cells is not sufficient. It is necessary to perform the Franz cell experiment using the ex-vivo skin, which has the stratum corneum rather than the membranes.

We thank the reviewer for this suggestion, the Franz cell experiments were performed using also human stratum corneum-epidermis (SCE) membranes derived from surgical operations, accordingly. At this regard both methods (lines 171-177) as well as result sections (lines 484-486, 504-522) have been improved, Figure 3 was changed and Figure 5 added.

Point 1. The delivered amount (and % to total amount) in SC, dermis, and receiver regarding time should be shown.

Response 1. We thank the reviewer for this suggestion, Table S1 has been changed in order to include the information about the percentage of QT associated to SCE membranes under Franz cell diffusion study. In addition, in the revised version, Figure 5b reports the diffusion kinetics of QT through SCE membranes. In the natural membranes we employed the dermis was not present, nonetheless we appreciate the reviewer suggestion and we will perform further studies in future, as reported in the conclusion section at lined 648, 649.

Remarkably, the experiments performed by the natural membrane resulted in QT diffusion profiles very similar to those obtained by STRAT-M, suggesting the suitability of the synthetic membrane system for this kind of evaluation.

Point 2. The receptor solution is 50% ethanol, which itself is a penetration enhancer, and why 50% ethanol was used for the receptor.

Response 2. For diffusion experiments such as those performed by Franz cells, the mixture ethanol:water 50:50, v/v is usually used in the case of lipophilic molecules, such as quercetin, in order to assure sink conditions (lines 182-183), as reported in “GUIDANCE, D. Transdermal and Topical Delivery Systems - Product Development and Quality Considerations. Food Drug Adm. 2019, 1–25. FDA-SUPAC-SS. Guidance for Industry Non-sterile Semisolid Dosage Forms. Scale-up and Postapproval Changes: Chemistry, Manufacturing and Controls. In vitro Release Testing and In Vivo Bioequivalence Documentation. U.S. Dep. Heal. Hum. Serv. 1997, 19–24.”

Point 3. Please add the explanation about the delivery mechanism of nano-particles with region of  200-400 nm in average size through the stratum corneum.

Response 3. We thank the reviewer for this observation. We would like to underline that ethosomes and transethosomes are nanovesicular systems rather than nanoparticles. The peculiarity of ethosomes and transethosomes is their elasticity, that promote their penetration potential. They can be described as soft, malleable vesicles thanks to the presence of ethanol, as described in different studies (Touitou, E.; Dayan, N.; Bergelson, L.; Godin, B.; Eliaz, M. Ethosomes - Novel vesicular carriers for enhanced delivery: Characterization and skin penetration properties. J. Control. Release 2000, 65, 403–418, doi:10.1016/S0168-3659(99)00222-9. Abdulbaqi, I.M.; Darwis, Y.; Khan, N.A.K.; Assi, R.A.; Khan, A.A. Ethosomal nanocarriers: The impact of constituents and formulation techniques on ethosomal properties, in vivo studies, and clinical trials. Int. J. Nanomedicine 2016, 11, 2279–2304, doi:10.2147/IJN.S105016. Natsheh H, Vettorato E, T.E. Ethosomes for Dermal Administration of Natural Active Molecules. Curr Pharm Des. 2019, 25(21), 2338–2348, doi:10.2174/1381612825666190716095826. Ascenso, A.; Raposo, S.; Batista, C.; Cardoso, P.; Mendes, T.; Praça, F.G.; Bentley, M.V.L.B.; Simões, S. Development, characterization, and skin delivery studies of related ultradeformable vesicles: transfersomes, ethosomes, and transethosomes. Int. J. Nanomedicine 2015, 10, 5837–5851, doi:10.2147/IJN.S86186; Song, C.K.; Balakrishnan, P.; Shim, C.K.; Chung, S.J.; Chong, S.; Kim, D.D. A novel vesicular carrier, transethosome, for enhanced skin delivery of voriconazole: Characterization and in vitro/in vivo evaluation. Colloids Surfaces B Biointerfaces 2012, 92, 299–304, doi:10.1016/j.colsurfb.2011.12.004.). Recently we have also demonstrated the capability of ethosomes and transethosomes to pass intact within cells such as keratinocytes, possibly because the presence of ethanol in in ethosomes and transethosomes could induce disordering effects in the plasmalemma region making contact with the nanovesicles, loosening lipid packaging, thus allowing the passage of these malleable nanocarriers without the typical endosome formation (Costanzo, M.; Esposito, E.; Sguizzato, M.; Lacavalla, M.A.; Drechsler, M.; Valacchi, G.; Zancanaro, C.; Malatesta, M. Formulative study and intracellular fate evaluation of ethosomes and transethosomes for vitamin D3 delivery. Int. J. Mol. Sci. 2021, 22, doi:10.3390/ijms22105341). In the revised version a brief paragraph has been added (lines 359-365).

Reviewer 2 Report

The paper is well documented, the practical part detailed and the results clearly presented.

Author Response

The paper is well documented, the practical part detailed and the results clearly presented.

We thank the reviewer for his kind comments

Reviewer 3 Report

The article entitled Ethosomes and transethosomes as cutaneous delivery systems for quercetin: a preliminary study on melanoma cells, shown the results of the development and evaluation of these systems. The novelty of the article is not high or at least the authors did not show or highlight what is the novelty of the manuscript compared with other similar found in the bibliography.

Also, I recommend the following point.

I recommend to re-write the abstract because most of the information presented are a description of the methos employed but there is very few information of results.

Line 157. QT refers to the quercetin found in the filtrate?

Line 169. The receptor solution is ethanol: water (50:50 v/v?). The other component of the solution should be reported. this solution allows to keep sink conditions along the experiment?. The same in line 171 with ethanol solution 30:XX v/v

Section 2.9. Experimental details should be included, i.e. the amount of samples and reactive used, the positive or negative controls, etc.

Line 320, please include the meaning of POE.

Line 343: “The improvement of vesicle size exerted by QT could”. With improvement authors means increase? Why the authors consider that the increase of particles means an improvement of the formulation?

QT entrapment: authors stated that they use ultracentrifugation. Ultracentrifugation usually involved high centrifugal “g” forces. I think authors really used ultrafiltration technique, where ultrafiltration membranes are usually employed (as described in material and method section) and no so high centrifugal forces. Please clarify the technique and correct accordingly in all text.

Table 3. readability would improve if flux and diffusion coefficient appear in the release section. In addition, why the M-strat flux is not calculated?

Line 380-381. According to table 3 the antioxidant activity of SOL-QT (3.82) is higher than ETO0.9-QT and T-ETO0.9-QT. this is different from authors stated in line 380-381.

Line 384-387. The higher antioxidant activity of T-ETO 2.7 could be also related with the higher amount of phosphatidyl choline employed (there are double bounds in the structure). To confirm the hypothesis of the authors, they should test placebo (empty) vesicles of evaluate their intrinsic antioxidant activity (if it is present).

Table 4 is not a result, this table or its content should be included in material and method section.

Figure 3. panel C should be deleted, panel D is more informative. Keep both plots are redundant. Also, the term flux is usually employed at permeation experiments, however authors reported release experiment, then they should use release kinetics equations and not permeation equations. Also figure 3 b data suggest the adjustment to a non-straight line, maybe could be better explained by a first order release kinetics.

Section 3.7.2. Could the authors suggest a reason for the observed results?

Section 4. Conclusion section should describe the conclusion of the experimental, usually references are not included, and no discussion should be done in this section. I recommend to authors to re-write conclusion section.

Author Response

The article entitled Ethosomes and transethosomes as cutaneous delivery systems for quercetin: a preliminary study on melanoma cells, shown the results of the development and evaluation of these systems. The novelty of the article is not high or at least the authors did not show or highlight what is the novelty of the manuscript compared with other similar found in the bibliography.

We are sorry that the reviewer does not recognize the novelty of our work, anyway, at lines 85-88 the novelty of the study has been highlighted “Despite some researchers have explored the possibility to treat melanoma by oral administration of nano-particulate systems [31], the use of QT loaded ETO and T-ETO as topical approach in the adjuvant treatment of melanoma has not yet been proposed.” The same aspect was already underlined in the conclusion section.

Moreover, as highlighted in the manuscript, the choice of membranes employed for in vitro diffusion studies is a topic of concern (lines 528-530), few studies about the importance of the influence of the membranes have been published by now.

Also, I recommend the following point.

Point 1. I recommend to re-write the abstract because most of the information presented are a description of the methos employed but there is very few information of results.

Response 1. The abstract has been re-written accordingly.

Point 2. Line 157. QT refers to the quercetin found in the filtrate?

Response 2. QT refers to the amount of QT associated to the vesicles, thus retained by ETO and T-ETO, as stated in the phrase:

…where QT is the amount of drug retained by the vesicles and TQT is the total content of QT employed for ETO and T-ETO preparation. (lines 159-160)

Point 3. Line 169. The receptor solution is ethanol: water (50:50 v/v?). The other component of the solution should be reported. this solution allows to keep sink conditions along the experiment?. The same in line 171 with ethanol solution 30:XX v/v

Response 3 Yes, it is. At lines 168 and 182-185 the information has been improved (ethanol:water 50:50, v/v). For diffusion experiments, such as those performed by Franz cells, the mixture ethanol:water 50:50, v/v is usually used in the case of lipophilic molecules, such as quercetin, in order to assure sink conditions (GUIDANCE, D. Transdermal and Topical Delivery Systems - Product Development and Quality Considerations. Food Drug Adm. 2019, 1–25. FDA-SUPAC-SS. Guidance for Industry Non-sterile Semisolid Dosage Forms. Scale-up and Postapproval Changes: Chemistry, Manufacturing and Controls. In vitro Release Testing and In Vivo Bioequivalence Documentation. U.S. Dep. Heal. Hum. Serv. 1997, 19–24.) Of course the ethanol solution was ethanol:water 30:70, v/v, as specified at line 185 in the revised version of the manuscript.

Point 4. Section 2.9. Experimental details should be included, i.e. the amount of samples and reactive used, the positive or negative controls, etc.

Response 4. We thank the reviewer for this comment, the text of paragraph 2.9 has been modified accordingly (lines 222-239).

Point 5. Line 320, please include the meaning of POE.

Response 5. the meaning (polyoxyethylene oxide) has been included (line 337).

Point 6. Line 343: “The improvement of vesicle size exerted by QT could”. With improvement authors means increase? Why the authors consider that the increase of particles means an improvement of the formulation?

Response 6. We thank the reviewer, the term improvement was improperly employed. The term has been substituted with increase (line 352).

Point 7. QT entrapment: authors stated that they use ultracentrifugation. Ultracentrifugation usually involved high centrifugal “g” forces. I think authors really used ultrafiltration technique, where ultrafiltration membranes are usually employed (as described in material and method section) and no so high centrifugal forces. Please clarify the technique and correct accordingly in all text.

Response 7. We agree with the reviewer. The term ultrafiltration has been replaced in the whole manuscript accordingly. The technique has been further specified at paragraph 2.6.

Point 8. Table 3. readability would improve if flux and diffusion coefficient appear in the release section. In addition, why the M-strat flux is not calculated?

Response 8. We would like to underline that Franz cell systems allow to mimic the diffusion of drugs through natural membranes such as skin or mucosae, in particular in this study the method was employed to predict QT diffusion from the different vesicular systems once applied on the skin. Thus in the manuscript there isn’t a “release section”. The term “release” is more appropriate in the case of drug delivery systems designed for systemic administration ways, such as the parenteral or the oral ones, usually in these case a dialysis method could be better employed.

QT diffusion coefficients (reported in Table 4) are also depicted as histograms in Figure 4 (section “3.6. In vitro QT diffusion kinetics”)

With regard to STRAT-M, QT flux from this type of membrane was not calculated due to the obtained sigmodal profiles.  Indeed, in the case of NY and PTFE, the fluxes were calculated from the slope of the lines obtained from the accumulation curve (as described at paragraph 2.7, lines 191, 192). In the case of STRAT-M®, its tortuous structure resulted in anomalous QT diffusion profiles, preventing the possibility to calculate QT fluxes. This was stated in the revised version (lines 505-507).

Point 9. Line 380-381. According to table 3 the antioxidant activity of SOL-QT (3.82) is higher than ETO0.9-QT and T-ETO0.9-QT. this is different from authors stated in line 380-381.

Response 9. We thank the reviewer for the correct observation: the sentence has been modified in accordance with the suggestions (lines 398-400).

Point 10. Line 384-387. The higher antioxidant activity of T-ETO 2.7 could be also related with the higher amount of phosphatidyl choline employed (there are double bounds in the structure). To confirm the hypothesis of the authors, they should test placebo (empty) vesicles of evaluate their intrinsic antioxidant activity (if it is present).

Response 10. We thank the reviewer for the correct observation, the empty placebo systems have already been tested simultaneously with the formulations, without showing any antioxidant activity, supporting the hypothesis reported by the authors. A new sentence regarding these tests has been inserted in paragraph 3.4. (lines 388,389).

Point 11. Table 4 is not a result, this table or its content should be included in material and method section.

Response 11. In the revised version of the manuscript the Table is included in material and method section, new Table 1.

Point 12. Figure 3. panel C should be deleted, panel D is more informative. Keep both plots are redundant. Also, the term flux is usually employed at permeation experiments, however authors reported release experiment, then they should use release kinetics equations and not permeation equations. Also figure 3 b data suggest the adjustment to a non-straight line, maybe could be better explained by a first order release kinetics.

Response 12. We would like to underline that diffusion experiments were performed, rather than release experiments, as above reported in response 8. Franz cell system associated to synthetic or natural membranes is a well-known method employed to perform diffusion experiments (GUIDANCE, D. Transdermal and Topical Delivery Systems - Product Development and Quality Considerations. Food Drug Adm. 2019, 1–25. FDA-SUPAC-SS. Guidance for Industry Non-sterile Semisolid Dosage Forms. Scale-up and Postapproval Changes: Chemistry, Manufacturing and Controls. In vitro Release Testing and In Vivo Bioequivalence Documentation. U.S. Dep. Heal. Hum. Serv. 1997, 19–24. Ng, S.F.; Rouse, J.; Sanderson, D.; Eccleston, G. A Comparative study of transmembrane diffusion and permeation of ibuprofen across synthetic membranes using Franz diffusion cells. Pharmaceutics 2010, 2, 209–223, doi:10.3390/pharmaceutics2020209.)

We agree with the reviewer that in some cases diffusion can follow a non-straight line. At this regard, in order to properly determine the flux in the case of NY and PTFE, the linear portion of the curve (within 5 h) has been considered in the revised version and included in supplementary data (Figure S1) (lines 457, 458). The F and D values were accordingly changed in Table 3 and Figure 4.

In the revised version, Figure 3 refers only to QT diffusion kinetics through NY and PTFE. Figure 5 has been added, showing the STRAT-M and stratum corneum-epidermis QT diffusion kinetics.

In the case of STRAT-M and SCE, due to the peculiar structure of the membranes, the profiles were characterized by a lag time and a sigmoidal profile, as described in the manuscript and documented in different papers (Haq, A.; Goodyear, B.; Ameen, D.; Joshi, V.; Michniak-Kohn, B. Strat-M® synthetic membrane: Permeability comparison to human cadaver skin. Int. J. Pharm. 2018, 547, 432–437, doi:10.1016/j.ijpharm.2018.06.012.; Sguizzato, M.; Valacchi, G.; Pecorelli, A.; Boldrini, P.; Simelière, F.; Huang, N.; Cortesi, R.; Esposito, E. Gallic acid loaded poloxamer gel as new adjuvant strategy for melanoma: A preliminary study. Colloids Surfaces B Biointerfaces 2020, 185, 110613, doi:10.1016/j.colsurfb.2019.110613.). The irregular diffusion profiles of QT through these membranes prevented to calculate flux and diffusion coefficient values.

Point 13. Section 3.7.2. Could the authors suggest a reason for the observed results?

Response 13. The role of ROS in would healing has been well studied, over production of ROS inhibits wound healing, while a homeostatic concentration can improve this process (doi: 10.1016/j.bbagen.2008.01.006.  Epub 2008 Jan 18.) since ROS can play a second messenger role.  It is possible that the slow release of quercetin, due to its entrapment in ETO and T-ETO, prevents the formation of ROS (either directly or via the activation of NRF2) and this leads to a slower wound closure response. In addition, considering that NFkB is involved in would healing, NRF2 activation (by quercetin) could inhibits the activation of NFkB and delay cellular proliferation and the recovery from the scratch wound (doi: 10.1042/BST20150014). A paragraph has been added to section 3.7.2., accordingly (lines 582-589).

Point 14 Section 4. Conclusion section should describe the conclusion of the experimental, usually references are not included, and no discussion should be done in this section. I recommend to authors to re-write conclusion section.

Response 14. The Conclusion section has been re-written accordingly.

Author Response

The research paper by Ferrara et al. describes the ethosomes and transethosomes as drug carrier for topical delivery. The manuscript is well structured, the data fully supports all the statements. The manuscript and the results of the study are of relevance for scientists working in the field of controlled release, drug delivery and material sciences. Тhis research paper is a complete report and deserves being published as it is.

We thank the reviewer for his kind comments

Round 2
